# Design Study of a Round Window Piezoelectric Transducer for Active Middle Ear Implants

**DOI:** 10.3390/s21030946

**Published:** 2021-01-31

**Authors:** Dong Ho Shin

**Affiliations:** Institute of Biomedical Engineering Research, Kyungpook National University, 680 Gukchaebosang-ro, Jung-gu, Daegu 41944, Korea; swap9552@naver.com; Tel.: +82-53-427-5538

**Keywords:** round window piezoelectric transducer, active middle-ear implants, mechanical vibration analysis, sensorineural hearing loss, polydimethylsiloxane (PDMS)

## Abstract

This report describes the design of a new piezoelectric transducer for round window (RW)-driven middle ear implants. The transducer consists of a piezoelectric element, gold-coated copper bellows, silicone elastomer (polydimethylsiloxane, PDMS), metal cylinder (tungsten), and titanium housing. The piezoelectric element is fixed to the titanium housing and mechanical resonance is generated by the interaction of the bellows, PDMS, and tungsten cylinder. The dimensions of PDMS and the tungsten cylinder with output characteristics suitable for compensation of sensorineural hearing loss were derived by mechanical vibrational analysis (equivalent mechanical model and finite element analysis (FEA)). Based on the results of FEA, the RW piezoelectric transducer was implemented, and bench tests were performed under no-load conditions to confirm the output characteristics. The transducer generates an average displacement of 219.6 nm in the flat band (0.1–1 kHz); the resonance frequency is 2.3 kHz. To evaluate the output characteristics, the response was compared to that of an earlier transducer. When driven by the same voltage (6 V_p_), the flat band displacement averaged 30 nm larger than that of the other transducer, and no anti-resonance was noted. Therefore, we expect that the new transducer can serve as an output device for hearing aids, and that it will improve speech recognition and treat high-frequency sensorineural hearing loss more effectively.

## 1. Introduction

Various types of middle-ear implant transducers [1,2,3,4,5,6,7,8,9,10] that stimulate round window (RW) have been developed [11,12,13,14,15]. However, there is no highly stable and practical RW transducer for middle-ear implants except for the floating mass transducer (FMT) developed by MED-EL Inc. (Innsbruck, Austria) [16,17]. FMT has a floating mass type structure. Since the vibration energy spreads to each side of the FMT, it cannot transmit all to the RW. In addition, according to a previously published case report, output characteristics are poor in the low-frequency region (below 1 kHz) [18,19]. In order to improve the output characteristics of the low-frequency band, it is important to develop a RW transducer with a new structure.

Shin and Cho developed a prototype RW transducer by combining a piezoelectric element, cantilever membrane, and bellows [20]. This RW transducer produced mechanical resonance via the cantilever membrane and bellows, so that high output was generated in the mid-frequency band (2–3 kHz). Here, the piezoelectric element was “floated”, i.e., was not fixed to the housing, for use as a mass. Temporal bone experiments showed high output in the low-frequency band. However, the frequency characteristics of the RW transducer of Shin and Cho have resonance and antiresonance. Resonance can deliver high output (energy) at a particular frequency, while antiresonance can drop output to almost zero at a particular frequency. Antiresonance can be a problem for hearing aids, which are deigned to transmit sound information. Therefore, there is a need for an improved transducer that can generate mechanical resonance without antiresonance.

This study designed and implemented an RW transducer with frequency characteristics that are better than the RW transducer of Shin and Cho [20]. The RW piezoelectric transducer compensates for sensorineural hearing loss, as did the earlier transducer. To transmit the vibration force generated by the piezoelectric element optimally to the RW, we also employ a very low-rigidity bellows as the vibration membrane. However, unlike previous transducers that “floated” in the sense that one side of the piezoelectric element was not completely fixed, the proposed transducer generates vibration in only one direction because one side of the piezoelectric element is fixed to the bottom of the titanium housing. This solves a major problem of the earlier transducer: the anti-resonance-mediated near-total loss of vibration in a required frequency band. Interactions between the metal cylinder, the synthetic bellows spring, and the silicone elastomer create mechanical resonance that generates a high output over the required frequency band. To identify a frequency appropriate for sensorineural hearing loss compensation, we subjected the dimensions of the silicon elastomer and the mass of the metal cylinder to finite element analysis (FEA) and used the data to build our transducer. A bench test confirmed the desired frequency characteristics. Finally, by comparing the bench test results with the frequency characteristics of the previous transducer, it was confirmed that the frequency characteristics of the proposed transducer were improved.

## 2. Piezoelectric Transducer Design

### 2.1. RW Drive Piezoelectric Transducer

The proposed transducer stimulates the round window membrane (RWM) by fixing the bottom or side of the housing to the RW niche wall (Figure 1). The vibrational energy generated from the transducer is transmitted in the RWM direction without spreading to the surrounding area.

The piezoelectric transducer consists of a bellows, metal cylinder, silicone elastomer, piezoelectric element, and titanium housing, as shown in Figure 2a. Figure 2b shows a cross-section of the piezoelectric transducer. The piezoelectric element used in this study was tensioned and compressed in the vertical direction when a voltage was applied by the converse piezoelectric effect. The bottom of the piezoelectric element was firmly fixed to the titanium housing, such that all vibrations generated by tension and compression were directed toward the vibration membrane. In addition, the spring rate of the silicone elastomer, mass of the metal, and spring rate of the bellows were combined to create mechanical resonance.

### 2.2. Mechanical Vibration Analysis

The proposed piezoelectric transducer can be expressed as an equivalent mechanical model consisting of two springs and one mass; its physical motion system is shown in Figure 3a. The physical system can be expressed as a mechanical vibration system composed of a mass (*m*), two springs (*k_B_* and *k_S_*), and two dampers (*c_B_* and *c_S_*) (Figure 3b) [21].

The mechanical vibration driven by an external force is:(1)$$m\ddot{x}+\left(c_{S}+c_{B}\right)\dot{x}+\left(k_{S}+k_{B}\right)x\;=\;F_{0}cos\omega t$$
where *m* is the mass of the metal cylinder, *c_B_* and *c_S_* are the air viscous damping factors of the bellows and silicone elastomer, respectively, *k_B_* and *k_S_* are the spring constants of the bellows and silicone elastomer, respectively, *F*_0_ is the vibrational force generated by the piezoelectric element when voltage is applied, and *ω* is the angular frequency. As the vibrational force and the solution are harmonic and have the same frequency, the solution can be written:(2)$$x_{p}\left(t\right)\;=\;Xcos\omega$$
where *X* is the maximum amplitude of *x_p_*(*t*). The homogeneous solution of Equation (1) is:(3)x(t) = Acosωt+Bsinωt

The velocity is the first derivative of x(t) and the acceleration the second derivative:(4)$$\dot{x}\left(t\right)\;=\;-\;A\omega sin\omega t+B\omega cos\omega t$$
(5)$$\ddot{x}\left(t\right)\;=\;-\;A\omega^{2}cos\omega t-B\omega^{2}sin\omega t$$

Substituting these into Equation (1), we obtain:(6)$$m\left(-\;A\omega^{2}cos\omega t-B\omega^{2}sin\omega t\right)+\left(c_{S}+c_{B}\right)\left(-\;A\omega sin\omega t+B\omega cos\omega t\right)\;+\left(k_{S}+k_{B}\right)\left(Acos\omega t+Bsin\omega t\right)=\;F_{0}cos\omega t$$
(7)$$\left[\begin{array}{cc} \left(k_{S}+k_{B}\right)-m\omega^{2} & \left(c_{S}+c_{B}\right)\omega^{2}\\ -\left(c_{S}+c_{B}\right)\omega^{2} & \left(k_{S}+k_{B}\right)-m\omega^{2} \end{array}\right]\cdot \left[\begin{array}{c} A\\ B \end{array}\right]=\left[\begin{array}{c} F\\ 0 \end{array}\right]$$

When Equation (7) is subjected to Cramer’s rule, *A* and *B* become:(8)$$A=\left[F\left\{\left(k_{S}+k_{B}\right)-m\omega^{2}\right\}\right]/\left[{\left\{\left(k_{S}+k_{B}\right)-m\omega^{2}\right\}}^{2}+{\left\{\left(c_{S}+c_{B}\right)\omega \right\}}^{2}\right]$$
(9)$$B=\left\{F\left(c_{S}+c_{B}\right)\omega \right\}/\left[{\left\{\left(k_{S}+k_{B}\right)-m\omega^{2}\right\}}^{2}+{\left\{\left(c_{S}+c_{B}\right)\omega \right\}}^{2}\right]$$

The displacement amplitude (*X*) of the piezoelectric transducer is:(10)$$X\;=F_{0}/\sqrt{{\left\{\left(k_{S}+k_{B}\right)-m\omega^{2}\right\}}^{2}+{\left\{\left(c_{S}+c_{B}\right)\omega \right\}}^{2}}\;[m]$$

The theoretical vibration displacement of the piezoelectric transducer can be derived using the above equations. However, it is complicated to perform sweep frequency response analysis and various parameter analyses using the motion equations and difficult to obtain an accurate value. In addition, it is necessary to analyze the motion correlations between the components of the transducer. Therefore, to confirm the optimal parameter values and motion correlation of the transducer components, FEA is required.

Generally, middle ear implants are suitable for patients with mild–moderate mixed or conductive hearing loss, or sensorineural hearing loss [22]. The transducer proposed in this study is an output device for hearing aids focused on compensation of sensorineural hearing loss. Middle ear implants provide superior speech intelligibility compared to other types of hearing aids [23,24,25]. Studies on the importance of different frequency bands to speech intelligibility have indicated that the maximum contribution occurs in the frequency band around 2.5 kHz [26]. In addition, according to audiograms in the case of sensorineural hearing loss, the hearing level drops sharply at frequencies above 1 kHz. To compensate for sensorineural hearing loss and provide excellent speech intelligibility, amplification is required in the 2–3 kHz frequency band. In addition, the transducer for an RW-drive middle ear implant reported previously by Shin et al. is similar to the transducer proposed in this paper, and it generates a resonant frequency at 2.2 kHz [27,28]. The transducer proposed in this paper aims to generate mechanical resonance around 2.2 kHz. The resonance frequency of the piezoelectric transducer is determined by the mass of the metal cylinder (*m*) and the sum of the spring constants of the bellows and silicone elastomer (*k_B_* + *k_S_*, respectively). That is, if the mass of the metal cylinder and spring constant of the bellows and silicon elastomer are combined appropriately, the resonant frequency targeted in this study can be achieved.

FEA was performed using COMSOL Multiphysics 5.4 software (COMSOL Inc., Burlington, MA, USA) to derive the output characteristics and resonance frequency of the proposed transducer as follows. The bellows (outer diameter 1.75 mm, inner diameter 1.21 mm, thickness 0.0076 mm, height without trim 0.5 mm; three corrugations) and piezoelectric element (width 0.9 mm, depth 0.9 mm, height 1.6 mm) had the same shape and physical properties as in the previous piezoelectric transducer [20]. Polydimethylsiloxane (PDMS), which is both biocompatible and biostable, served as the silicon elastomer. PDMS with various mechanical characteristics can be prepared by adjusting the proportions of the base elastomer and the curing agent. In addition, the viscosity is low; the material is easily injected into a very small mold. Tungsten, which has high specific gravity, was selected as the metal cylinder. A metal with high specific gravity was used because a weight change is evident even if the size of the metal cylinder changes only slightly. There are metals with a higher specific gravity than tungsten (specific gravity 19.3), but they are not easy to obtain and are expensive. Therefore, tungsten was used, as it is relatively cheap and can be obtained easily. A three-dimensional (3D) mesh model of the transducer was constructed through the Solid Mechanics interface of the COMSOL Structural Mechanics Module (Figure 4a). The titanium housing, which is not involved in the vibration characteristics of the transducer, was excluded from the model. The vibration displacement of the piezoelectric element was set to generate a flat displacement of 320 nm (from 0.1 to 10 kHz) when 6 V_P_ was applied using the Electrostatics interface. Then, the Solid Mechanics and Electrostatic interfaces were coupled using the Piezoelectric effect in the Multiphysics interface. The material properties of the bellows, tungsten, PDMS, and piezoelectric element (using the PZT-5H material built into COMSOL software) used in the FEA had the following features: bellows density, 8960 kg/m^3^; bellows Poisson’s ratio, 0.355; bellows Young’s modulus, 119 × 10^9^ N/m^2^; tungsten density, 19,300 kg/m^3^; tungsten Poisson’s ratio, 0.27; tungsten Young’s modulus, 340 × 10^9^ N/m^2^; PDMS density, 970 kg/m^3^; PDMS Poisson’s ratio, 0.49; PDMS Young’s modulus, 750 × 10^3^ N/m^2^. The bellows trim and bottom of the piezoelectric element were applied in “fixed constraint” mode and the “free tetrahedral” type was applied to the transducer 3D mesh model (253,347 domain elements; 32,682 boundary elements; 4504 edge elements; maximum element size, 0.311 mm; minimum element size, 0.0559 mm; maximum element growth rate, 1.5; curvature factor, 0.6; and narrow-region resolution, 0.5).

The spring rate of the bellows was derived using Equation (11), and the calculated value was 2059.48 N/m. The critical damping coefficient of the bellows, calculated using Equation (12), was 0.287 N∙s/m. The damping ratio is 0.09, which is derived from the critical damping coefficient [28]. And the damping ratio of the PMDS was 0.1 [29]. To apply the energy loss of the transducer by the bellows and PDMS, it was set using the damping of the Solid Mechanics interface; the damping type is Rayleigh damping and the input parameter is the damping ratio. To calculate all dependent variables applied to the model at the same time, the Suggested Direct-Fully Coupled method of Stationary Solver and Eigenvalue Solver were used as the analysis type. The average value of the linear error is 3.73 × 10^−11^. The factor in error estimate is set to 1.
(11)$$k=1.7\left(D_{m}E_{b}t_{p}^{3}n/w^{3}C_{f}N\right)$$
(12)$$C_{c}=2\sqrt{km}$$
where *k* is the spring rate of the bellows, *D_m_* is the outer diameter of the bellows, *E_b_* is Young’s modulus of elasticity for the bellows material, *t_p_* is the bellows material thickness for one ply, *ω* is the corrugation depth, *C_f_* is the correction factor for the bellows’ stiffness (from the curve in Figure 4.17 in the EJMA Standards), *n* is number of the bellows’ material plies of thickness, *N* is the number of corrugations active in the bellows, *C_c_* is the critical damping of the bellows, and m is the mass of the bellows [28].

The parametric elements used in the analysis controlled only the diameter and height of the PDMS and the mass of tungsten. In addition, the height of the proposed transducer was fixed at 3.2 mm based on the anatomical structure of the RW niche [30], which is the implantation location. Therefore, the combined total height of PDMS and tungsten could not exceed 1.5 mm (total height of the piezoelectric element, housing bottom, and bellows, 1.7 mm). To determine the dimensions of PDMS, the diameter of tungsten was fixed at 1.12 mm, and the analysis was performed while changing the height (from 0.1 to 0.6 mm in increments of 0.1 mm) and diameter (from 0.3 to 1.0 mm in increments of 0.1 mm) of the PDMS. Here, the diameter of tungsten was set as the maximum in consideration of the inner diameter of the bellows, and the mass (volume × specific gravity) of tungsten was calculated numerically, reflecting how the height of tungsten changes with that of the PDMS. Figure 4b shows the total displacement (based on static analysis), which is the result of the FEA, and Figure 5a–f show graphs of frequency–response characteristics (based on dynamic analysis) according to the variable elements of PDMS and tungsten. FEA showed that the greater the PFMS height and the smaller the diameter, the greater the displacement loss from the piezoelectric element to the bellows. Moreover, the smaller the PDMS height, the more clearly the resonance frequency shifted with PDMS diameter. In other words, as the PDMS height was reduced, it became easier to adjust the resonance frequency while reducing the displacement loss. The green line (2.3 kHz) in Figure 5a, and the blue and pink lines (2.1 kHz and 2.3 kHz, respectively) in Figure 5b, are graphs of high displacement in the flat region (below 1 kHz) that satisfy the resonant frequencies targeted in this study. Among them, the green line in Figure 5a was finally selected because the output size of the flat region of the green line (average, 203.2 nm) was larger than those of the blue (average, 108.5 nm) and pink (average, 141.7 nm) lines in Figure 5b. The green line in Figure 5a corresponds to PDMS with a diameter of 0.5 mm and height of 0.1 mm, and tungsten with a diameter of 1.12 mm, height of 1.4 mm, and mass of 26.6 mg.

## 3. Piezoelectric Transducer Implementation

### 3.1. Fabrication of Transducer

Each component of the RW piezoelectric transducer was manufactured based on parameters derived from the FEA results. Among the components, the piezoelectric elements (PAZ-10-0079; Murata Manufacturing Co. Ltd., Kyoto, Japan) and bellows (custom-made by Servometer Inc., Cedar Grove, NJ, USA) were also used in previous studies. The piezoelectric element had a width of 0.9 mm, depth of 0.9 mm, and height of 1.6 mm, and the bellows had a diameter of 1.75 mm, height of 0.8 mm (including the trim height of 0.3 mm), and three corrugations. The titanium housing (Ti-6AL-4V ELI; 1.75 mm in diameter and 2.5 mm in height) and tungsten (1.21 mm in diameter, 1.4 mm in height, and 26.9 mg in mass; 99.95% pure tungsten) were machined by computer numerically controlled (CNC) lathe milling. PDMS was manufactured using Sylgard^®^184 (two-part polymer: base elastomer and curing agent; Dow Corning, Midland, MI, USA). After mixing the two parts according to the standard mixing ratio (10 parts base elastomer and 1 part curing agent) of PDMS [31], air bubbles were completely removed in a vacuum chamber. The elastomer without bubbles was added to a mold (Figure 6), and solidified at 100 °C for 40 min in a temperature-controlled chamber. The hardened elastomer was then removed from the mold to obtain PDMS with a diameter of 0.5 mm and height of 0.1 mm (Figure 6). Figure 7a shows an exploded view of the RW piezoelectric transducer parts. The transducer was assembled at a probe station equipped with a microscope, and the assembled transducer is shown in Figure 7b.

### 3.2. Transducer Output Characteristics

The output characteristics of the piezoelectric transducer were measured using a non-contact vibration measurements system consisting of a data acquisition unit (NI PXIe-8840 controller, NI PXIe-1071 chassis and NI PXI-4461 sound-vibration module; National Instruments Corp., Austin, TX, USA) and a single-point laser Doppler vibrometer (LDV; OFV-551 sensor head and OFV-5000 controller; Polytec GmbH, Waldbronn, Germany) [32]. The measurement system generated a sinusoidal signal to drive the piezoelectric transducer, and simultaneously collected the vibration signals of the piezoelectric transducer measured by the LDV (Figure 8a). Figure 8b shows an experimental environment for measuring the output characteristics of the proposed piezoelectric transducer. The transducer in the no-load condition was firmly fixed to the anti-vibration table using cyanoacrylate glue, and the angle of incidence of the laser beam was adjusted to be perpendicular to the center of the bellows using a micromanipulator joystick (A-HLV-MM30; Polytec GmbH). After applying a constant voltage of 6 V_P_ to the transducer through the sound-vibration module (NI PXI-4461), the output characteristics of the transducer were measured with LDV. The output characteristics of the proposed RW piezoelectric transducer (red line) measured by LDV are shown in Figure 9. A comparison of the measurement and simulation results (black line) indicated that the frequency-response characteristics were near-identical. The resonance frequencies were generated at the same location. However, slight differences in frequency magnitude were apparent, reflecting a mass error during tungsten processing and some misalignment during assembly. The tungsten mass difference (theoretical compared to actual) was 0.3 mg. However, as shown in Figure 9, this very small difference did not affect the location of transducer resonance. If the transducer were to be manufactured in a more advanced facility, differences in output characteristics caused by misalignment would be reduced.

## 4. Discussion and Summary

Here, this report describes a new RW-driving piezoelectric transducer with improved output characteristics; it also lacks the anti-resonance-caused output degradation of an earlier transducer [20]. The structure of the new and earlier transducer differs greatly; the piezoelectric element is now fixed rather than floating to eliminate anti-resonance. One side of the piezoelectric element is fixed to the bottom of the titanium housing. Vibrations are generated in only one direction, eliminating anti-resonance. Mechanical resonance is caused by interactions between the tungsten mass, the synthetic spring of the bellows, and the PDMS. FEA was used to explore various PDMS dimensions and the tungsten mass to optimize sensorineural hearing loss compensation. PDMS of diameter 0.5 mm and height 0.1 mm and a tungsten block of height 1.4 mm and a mass of 26.6 mg delivered the required frequency characteristics. The transducer was built and bench-tested under no-load conditions, confirming the desired frequency characteristics.

The responses were compared with those of the Shin and Cho transducer at the same driving voltage [20]. As shown in Figure 10, in the low frequency band (0.1–1 kHz), the output of the new transducer (red line) was larger than that of the earlier transducer (black line) by an average of 30 nm. The earlier “floating” transducer generated resonance at 2.4 kHz but this fell sharply at 3.2 kHz because of anti-resonance. Our new transducer generated resonance at 2.3 kHz and then smoothly reduced the output without anti-resonance. In other words, in the 2–3 kHz band, which compensates for sensorineural hearing loss and is essential if speech is to be intelligible, the new transducer was better than the early transducer. We expect that the new transducer can serve as an output device for hearing aids that improve speech recognition and treat high-frequency sensorineural hearing loss.

To determine whether the proposed transducer is suitable for use as an RW-drive middle ear transducer, it will be necessary to conduct further studies of the transducer output characteristics in an implanted environment. Further nonclinical validation studies are required to examine the vibration transmission characteristics of the proposed transducer through temporal bone or animal experiments.

## Figures and Tables

**Figure 1 sensors-21-00946-f001:**
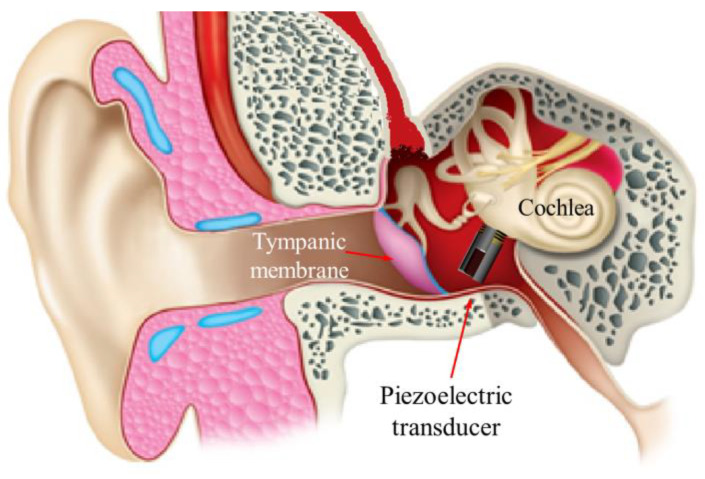
Schematic diagram of a piezoelectric transducer installed in the round window (RW).

**Figure 2 sensors-21-00946-f002:**
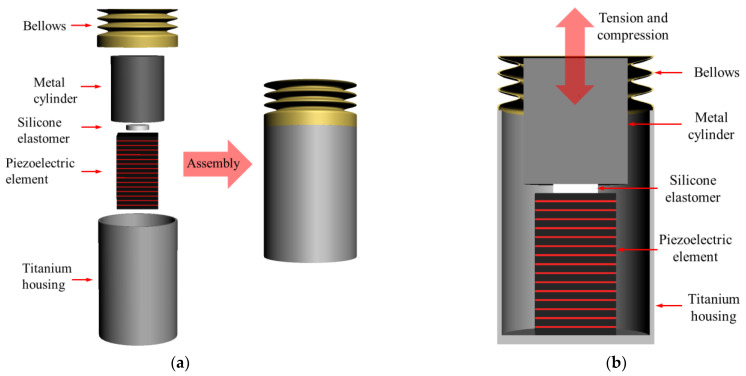
(**a**) Components of the piezoelectric transducer; (**b**) cross-section of the piezoelectric transducer.

**Figure 3 sensors-21-00946-f003:**
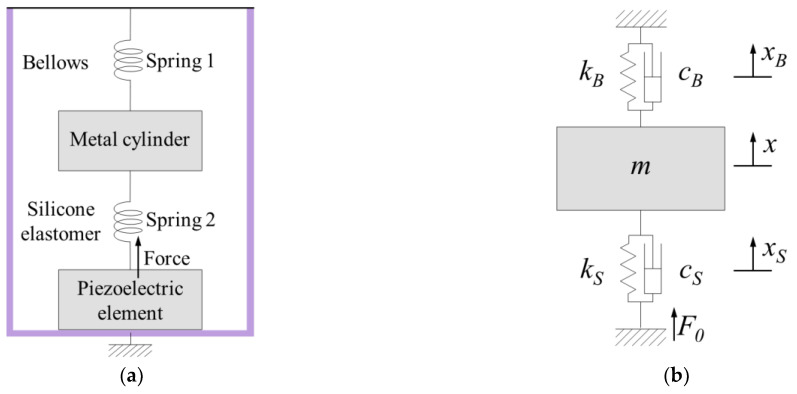
(**a**) Physical system of a piezoelectric transducer; (**b**) simplified model with the spring mass damper system of a piezoelectric transducer [21].

**Figure 4 sensors-21-00946-f004:**
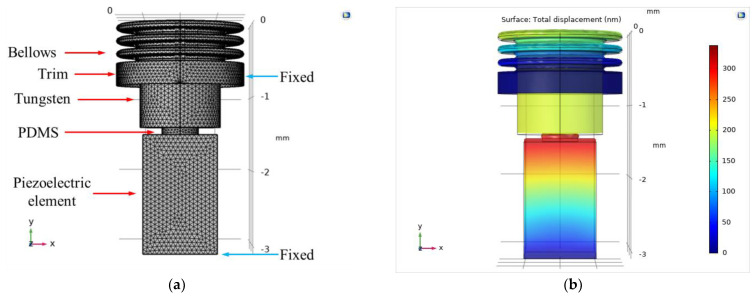
(**a**) 3D finite element analysis (FEA) mesh model of the RW piezoelectric transducer; (**b**) the FEA results.

**Figure 5 sensors-21-00946-f005:**
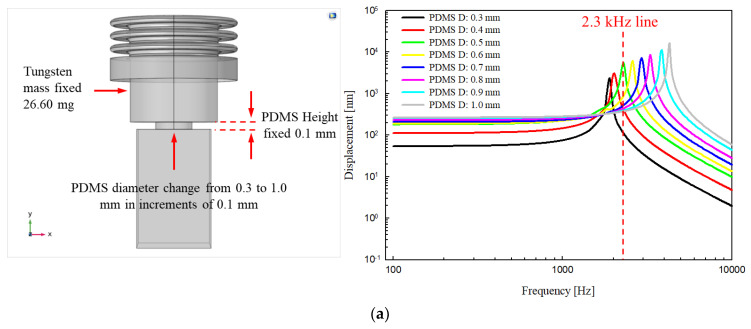
Displacement frequency–response characteristics according to the variable elements of polydimethylsiloxane (PDMS) and tungsten. Fixed parameters: (**a**) PDMS height, 0.1 mm; tungsten mass, 26.60 mg; (**b**) PDMS height, 0.2 mm; tungsten mass, 24.70 mg; (**c**) PDMS height, 0.3 mm; tungsten mass, 22.80 mg; (**d**) PDMS height, 0.4 mm; tungsten mass, 20.90 mg; (**e**) PDMS height, 0.5 mm; tungsten mass, 19.00 mg; (**f**) PDMS height, 0.6 mm; tungsten mass, 17.10 mg.

**Figure 6 sensors-21-00946-f006:**
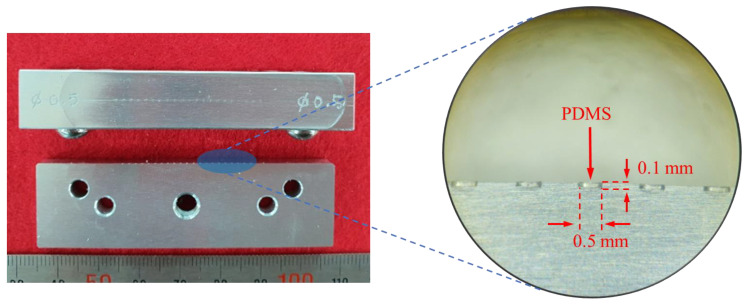
Mold for manufacturing PDMS (**left**) and manufactured PDMS (**right**).

**Figure 7 sensors-21-00946-f007:**
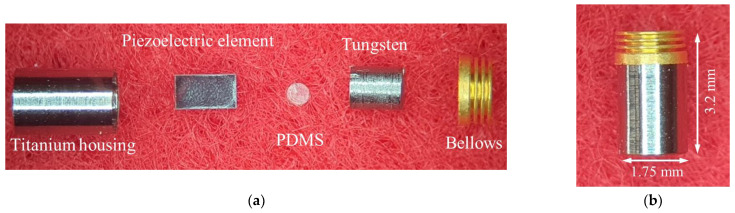
(**a**) Exploded view of the proposed RW piezoelectric transducer; (**b**) Assembled transducer.

**Figure 8 sensors-21-00946-f008:**
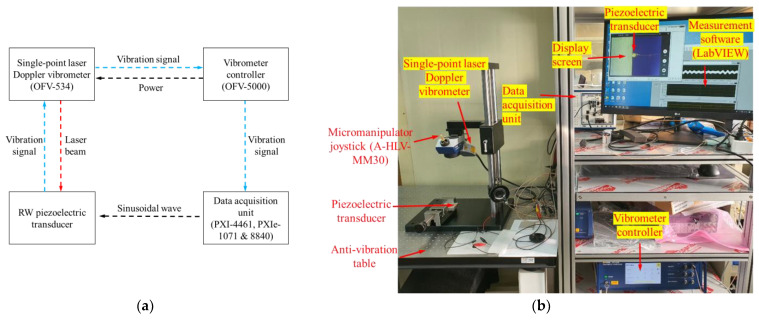
(**a**) Block diagram of the measurement system; (**b**) experimental environment for measuring piezoelectric transducer output characteristics.

**Figure 9 sensors-21-00946-f009:**
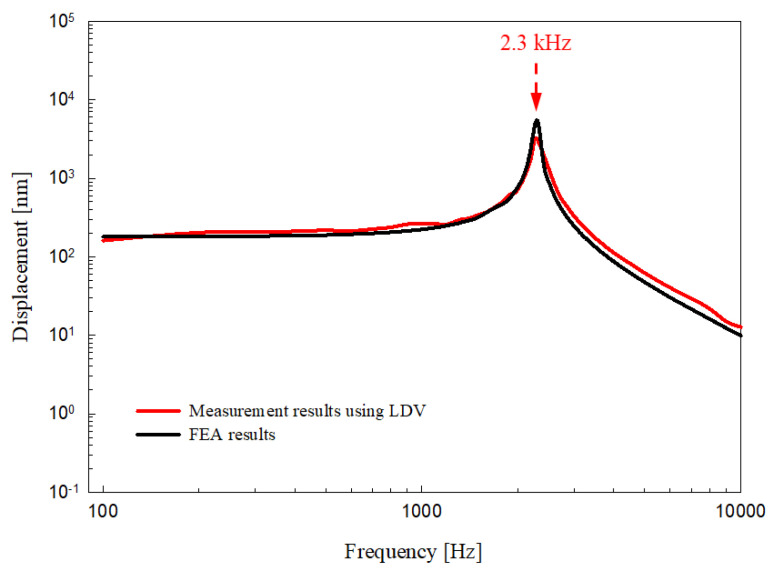
Comparison of output characteristics between the FEA model and RW piezoelectric transducer.

**Figure 10 sensors-21-00946-f010:**
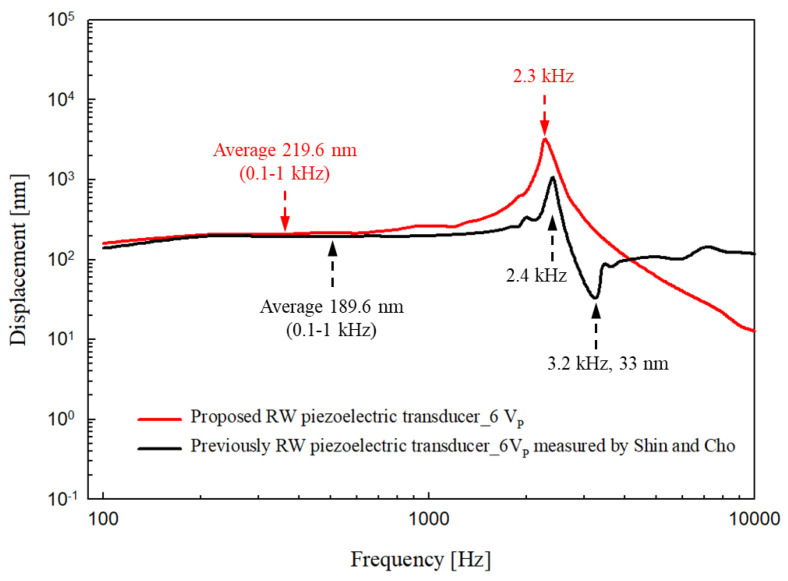
Comparison of the response tendencies of the proposed piezoelectric transducer (red line) and the previous one (from Shin and Cho; black line) driven by 6 V_P_.

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
