# Peer review of "Design Study of a Round Window Piezoelectric Transducer for Active Middle Ear Implants"

_sensors, 2021, doi:10.3390/s21030946_

Round 1

Reviewer 1 Report

There are critical comments for the submitted manuscript.

1. In the abstract, authors does not show the important measured data with proposed important concept.
2. Author proposed very similar fabrication methods and results compared to previous publications. In addition, pictures are similar with previous one. It is hard to recognize the novelty of the proposed piezoelectric transducers using PDMS materials. For piezoelectric transducers, PDMS has been widely used due to soft characteristics and low electromechanical factor. Therefore, authors need to compare the proposed transducers using PDMS materials and other materials.
3. Author only submitted the article with 9 pages so the article should be communication. The novelty of the article is very important for the communication. However, the proposed round window piezoelectric transducer for ear implants does not show such a novel idea compared to the previous article. Thus, author needs to extend the article more than 15 pages for full article. Otherwise, the manuscript cannot be published with current form of the paper.
4.Author used PDMS materials for piezoelectric transducers for proposed ear applications. However, authors need to emphasize the different characteristics compared with other materials in the introduction sections with various references.

There are minor comments to be updated.
1. Quality of Figures are too low. Author need to improve the quality.
2. Labels and letters in all Figures are too small.
3. Author does not mark important measured data in some Figures.
4. Author contributions have wrong format for MDPI.
5. Funding and acknowledgments are almost same.
6. Author needs to use abbreviated journal name.
7. Author needs to provide location information for the textbook.
8. Author needs to provide the reference for Figure 3.

Reviewer 2 Report

Report on paper "Design study of round window piezoelectric transducer for active middle ear implants" submitted by Shin, for publication in Sensors (sensors-1071913).

The author implemented a round window piezoelectric transducer based on finite element analysis and the fabricated device was experimentally tested under no-load conditions to confirm the output characteristics and compared to another transducer in term of specifications. Even though the topic of the paper is interesting, the author should highlight the originality of this work with respect to reference 10. In view of the presented results, some discussions of the manuscript are lacking depth. The paper cannot be accepted in its present form and the authors must perform substantial modifications by addressing the following comments:

  1. In the abstract, the transducer performances should be evaluated.
  2. The literature survey in the introduction lacks of references in the field of MEMS transducers for middle ear implants, which is a topic deeply investigated in the recent past.
  3. In the introduction, the originality of the paper should be clearly highlighted with respect to the previous author work in reference 10.
  4. The overlap with reference 10 (including figures) should be significantly reduced.
  5. The damping in equation (1) cannot be completely neglected; otherwise the displacement at resonance in equation (5) will be infinite.
  6. In section 2, full details about the FE model (type of elements, mode size, solver, convergence, stability…) should be given.
  7. Some frequency responses in figure 5 are not sufficiently smooth which can be due to the frequency step size that should be refined close to the resonance peak.
  8. The difference between the FE and experimental results in Figure 9 should be explained or justified.
  9. The author should explain how the damping was implemented in the FE model. Is it the measured damping? How it was measured? Which kind of damping has been used?
  10. Section 4 must be extended. The description of the results lacks of depth and the authors should provide a detailed explanation of each result according to the physics phenomena and the significance of each result for active middle ear implants.
  11. The performance of the proposed transducer could be compared with respect to other devices from the literature and not only with respect to the previous author work.
  12. The quality of the figures should be enhanced.

Reviewer 3 Report

The manuscript describes a design study of round window piezoelectric transducer towards application in ear implants. The transducer consists of a piezoelectric element, gold-coated copper bellows, silicone elastomer (polydimethylsiloxane, PDMS), metal cylinder (tungsten), and titanium housing. The study used equivalent mechanical model and finite element analysis in selection for the dimensions of PDMS and the tungsten cylinder. In addition, bench tests were performed under no-load conditions to confirm the output characteristics of the piezoelectric transducer.

Overall, this is a good work and could be suitable for publication in the current journal. However, I suggest the following points to be fully addressed before publication

  1. The quality of Figures and images should be improved. Especially, the font size is small in some figures, making them difficult to read
  2. Discussion/model should be given to understand the shift of the resonant frequency with PDMS dimension in Figure 5. Did the authors consider the thickness of PDMS?
  3. There is no demonstration for an implanted environment or temporal bone or animal experiments. The author needs to remove the claim on the suitability of the device for speech recognition and treating high-frequency sensorineural hearing loss.

Round 2

Reviewer 1 Report

Authors corrected all of the questions and updated the novelty of the proposed concept so the paper can be accepted.

Reviewer 2 Report

The authors have addressed my comments sufficiently to recommend publication of the paper in its current form.